# Effects of Accelerating the Ageing of 1D PLA Filaments after Fused Filament Fabrication

**DOI:** 10.3390/polym15010069

**Published:** 2022-12-24

**Authors:** Jaime Orellana-Barrasa, Sandra Tarancón, José Ygnacio Pastor

**Affiliations:** Centro de Investigación en Materiales Estructurales (CIME), Universidad Politécnica de Madrid, 28040 Madrid, Spain

**Keywords:** 1D PLA filaments, mechanical properties, thermal properties, temperature, ageing

## Abstract

The effects of post-treatment temperature-based methods for accelerating the ageing of PLA were studied on 1D single-PLA filaments after fused filament fabrication (FFF). The goal was to answer the questions whether the PLA can be safely aged—i.e., without degrading—at higher temperatures; at which temperatures, if any; how long it takes for the PLA to fully age at the chosen temperature; and which are the main differences between the material aged at room temperature and the material aged at higher temperatures. We also share other helpful information found. The use of 1D filaments allows for decoupling the variables related to the 3D structure (layer height, raster angle, infill density, and layers adhesion) from the variables solely related to the material (here, we analysed the molecular weight, the molecular orientation, and the crystallinity). 1D PLA filaments were aged at 20, 39, 42, 51, 65, 75, and 80 °C in a water-bath-inspired process in which the hydrolytic degradation of the PLA was minimised for the ageing temperatures of interest. Those temperatures were selected based on a differential scanning calorimetry (DSC) scan of the PLA right after it was printed in order to study the most effective ageing temperature, 39 °C, and highlight possible degradation mechanisms during ageing. The evolution of the thermal and mechanical properties of the PLA filaments at different temperatures was recorded and compared with those of the material aged at room temperature. A DSC scan was used to evaluate the thermal and physical properties, in which the glass transition, enthalpic relaxation, crystallisation, and melting reactions were analysed. A double glass transition was found, and its potential implications for the scientific community are discussed. Tensile tests were performed to evaluate the tensile strength and elastic modulus. The flow-induced molecular orientation, the degradation, the logistic fitting, and the so-called summer effect—the stabilisation of properties at higher values when aged at higher temperatures—are discussed to assess the safety of accelerating the ageing rate and the differences between the materials aged at different temperatures. It was found that the PLA aged at 39 °C (1) reached almost stable properties with just one day of ageing, i.e., the ageing rate accelerated by 875% for the elastic modulus and by 1635% for the yield strength; (2) the stable properties were higher than those from the PLA aged at room temperature; and (3) no signs of degradation were identified for the ageing temperature of interest.

## 1. Introduction

Additive manufacturing is a growing field that can produce 3D-printed components with few geometrical restrictions. These manufacturing processes are suitable for polymers, metals, ceramics, and composites [1,2,3,4,5,6]. Fused filament fabrication (FFF) is shared among the different additive manufacturing techniques due to its low cost and accessibility. The FFF works with materials that melt and flow (i.e., that are not suitable for thermosets), such as PLA [7,8,9], PETG [10], and ABS [11,12]. In this work, we study polylactic acid (PLA), a polymeric biomaterial commonly studied in many applications [13], from basic furniture and decoration [5,14] or textile and fashion [9] to advanced medical applications [7,15,16]. All of this is of great interest, as an accessible technique (FFF) is suitable for producing components with almost no restrictions with a relatively cheap material (PLA). However, before testing any PLA or PLA-based composite materials produced via FFF, it must be considered that FFF processing melts and extrudes the PLA. This creates a thermal treatment that returns the polymer to an out-of-equilibrium state from which it slowly ages, which in this work means an evolution of the material due to the thermal vibration of the macromolecules to their equilibrium position. Because of this, the properties of the printed PLA slowly evolve (age) and increase the elastic modulus, the tensile strength, the yield strength, the glass transition, the enthalpic relaxation, and any other property related to the conformation of the amorphous part of the PLA. Logically, and knowing this phenomenon, it should be mandatory to fully age the PLA—and explicitly prove that it is entirely aged—to make possible a reliable comparison of results between researchers when the effect of any variable is studied on the properties of printed PLA filaments or structures. That said, in our first work on PLA [17], we performed a comprehensive study on how the 1D filaments of printed neat PLA 2003D (one of the many PLA grades) naturally aged. We found that this high-molecular-weight PLA required up to 90 days—at room temperature, inside zip bags with a desiccant—to stabilise its mechanical and thermal properties and that these properties increased by up to 75% simply because of the ageing. An essential and argued aspect of that work was the use of 1D filaments instead of the standardised 3D-printed samples. Additive manufacturing via FFF consists in placing a continuous flow of fused material at particular locations, layer by layer, until the addition of material creates an object. The manner of producing these components layer by layer makes them highly anisotropic, and several variables need to be defined that have a significant influence on the properties of the final printed product: infill density, nozzle temperature, printing speed, layer thickness, raster angle, and bed temperature [18,19,20,21]. This inherent fact about any 3D-printed component makes understanding the material difficult using 3D-printed samples. It is almost impossible to separate the structure-related variables (the layer adhesion or the concentration of stresses on corners) and the material-related variables (the molecular weight, the crystallinity, and the orientation of the molecules). Surprisingly, no researchers, as far as we know, have studied the properties of 1D-printed PLA specimens by decoupling the material properties from the structural variables [22]. Nonetheless, studies on 3D-printed samples have highlighted the difficulties of addressing the multivariable problem of studying 3D-printed components [18].

Altogether, the facts from our previous work [18], in which the PLA 2003D required up to 90 days at room temperature to be entirely aged, raise the question of how to control the ageing rate of the printed PLA filaments and measure the real impact on the PLA. We performed a second work demonstrating that the PLA’s ageing can be safely stopped (without any decrease in the mechanical properties) by freezing the PLA at −24 °C in a conventional freezer [23]. However, we only studied lower temperatures—as our goal in that study was to completely stop the ageing—and did not conduct any research on higher temperatures for accelerating its ageing. In reviewing the literature, we could not find answers to our research questions posed here. Thus, we waited three months before making any characterisation of neat PLA material afterwards, at least until now (except for composite PLA-based materials). It is not new information that the ageing rate of a polymer can be accelerated by increasing the temperature. However, increasing temperatures might be problematic for PLA materials, as this enhances any potential degradation of the PLA, especially in humid atmospheres, based on the hydrolytic degradation due to the cleavage of ester groups [24,25,26,27,28,29]. This degradation should not be a problem at temperatures below the PLA’s glass transition temperature (T_g_) [19].

As the macromolecules gain more and more kinetic energy due to an increase in the temperature, they have more energy to reach more stable locations on the microstructure of the polymer, reducing its free space and ageing the material. For effective ageing (in which the properties increase until they stabilise), it would be ideal to use temperatures as high as possible but consistently below the T_g_. However, if temperatures above T_g_ are used, the macromolecules have so much kinetic energy that they can no longer be in ordered and stable positions—it is as though the amorphous part were to melt—and the ageing of the PLA is no longer possible.

The T_g_ is a second-order transition in which the polymer chains significantly increase their mobility. It can be measured using differential scanning calorimetry (DSC) or fast scanning calorimetry (FSC) as an endothermal drop related to the change in heat capacity [30], or through dynamic mechanical analysis (DMA) as a drop on the elastic modulus, as an increase in the loss modulus, and as an increase as well of the tan δ [31,32]. In this work, the DSC technique was used. The glass transition was not instantaneous. The material did not heat evenly during the DSC (depending on the mass of the material, the manner in which it is placed inside the DSC crucible, and the heating rate). The T_g_ also depends on a molecular weight distribution which, by its essence, makes the T_g_ into a distribution of temperatures. Even though the T_g_ is defined at some point between a changing line in the DSC graph (a transition), there is an onset and offset temperature. There are different theories for defining the specific T_g_ value from a DSC, but they all provide a T_g_ value higher than the onset temperature. Thus, lower values than the usually defined T_g_ should be used to ensure that all the PLA is ageing. More precisely, the ageing temperatures should be lower than the onset temperature measured at a slow heating rate and with as much surface of the PLA in direct contact with the DSC crucible as possible. This will ensure that all the amorphous part of the PLA (which, as a polymer, is formed through a weight distribution of macromolecules) remains restricted to low degrees of mobility, leading to an effective ageing process [24]. Otherwise, for a T_g_ value measured at 10 °C, ageing performed at 1 degree below that measured T_g_ might not age the polymer or, at least, not all of its structure.

Regarding the literature about ageing PLA, many researchers have studied the effect of heating the PLA after it was processed via FFF [19,33,34]. For example, some researchers have studied thermal treatments at temperatures above T_g_ to improve the mechanical properties of PLA by increasing the crystallinity [19,34]. Alternatively, other work mentions an improvement in mechanical properties due to the enhanced bond between the layers and an increase in crystallinity [35]. However, as far as we know, no research has considered the isolated 1D-printed filament but the whole 3D-printed structure after FFF [12,13,14,27,29].

This paper analyses and discusses the effects solely affecting the PLA when it is accelerated by raising the ageing temperature in a low-humidity atmosphere by studying those 1D filaments. Our research questions to answer are:Is it safe to effectively accelerate the ageing of the PLA by heating it? To answer this question, the degradation of the PLA was analysed with regard to the shifting of peaks in the DSC scans (decreased temperature in the crystallisation reactions, increase in the crystallisation enthalpies, a decrease in the melting point) and the decrease in both the elastic modulus and tensile strength. As the primary degradation mechanism is the hydrolytic degradation with the air humidity, the change in colour of a desiccant inside the ageing device was used to determine the low-humidity conditions during the ageing qualitatively.At which temperature can the PLA be effectively and safely accelerated? How long does the PLA need to be entirely aged at that temperature? Different temperatures were studied. The temperatures were chosen based on a DSC scan on the PLA right after it was printed after finding an optimal ageing temperature and forcing the degradation mechanism. The evolution of the thermal and mechanical properties was measured at different times to ensure that stable properties were reached and the minimum ageing time was set for the central ageing temperature of interest.What are the main differences between the PLA that was slowly aged at room temperature and the one aged at higher temperatures? The PLA aged at room temperature is compared with the material effectively aged at higher temperatures. This makes possible a better understanding of how future 3D-printed samples of PLA will be affected—at least for the thermal and mechanical properties of the 3D structures that are no longer slowly aged during 3 months at 20 °C but accelerated at a higher temperature.Has any other phenomenon of interest been detected? As different properties were studied, the results obtained during this research were analysed to determine other phenomena related to the PLA and, if something interesting is found, to share it with the research community.

To answer these questions and solely address the variables related to the material, 1D filaments were studied, following the methodology developed in previous research [15,17,23] based on the UNE-EN ISO 527-2:2012. The degradation, the flow-induced molecular orientation, the so-called summer effect, and other phenomena of interest (a hidden T_g_) are analysed. This paper complements our previous study on controlling the ageing of the PLA [23], in which the effects of freezing the PLA to stop its ageing were understood. The results of that paper are used in this research. Now, we are looking to understand the opposite, that is, how accelerating the ageing rate by increasing the ageing temperature impacts the mechanical and thermal properties of the filaments after FFF, the building block of any 3D-printed structure for answering those questions.

Answering our research questions will provide the following benefits:5.Our main aim is to find and understand a reliable, safe, and effective method to accelerate the ageing of PLA 4043D-based materials (decoupled from the effects of the structure) to be sure of how the ageing method is affecting the PLA and reduce the research schedule from months (time required until the PLA is entirely aged at room temperature) to just a few days (as demonstrated in this study).6.An emphasis on the importance of studying aged materials is essential for comparing results from different research with confidence.7.It would allow us to understand and report how the thermal treatments affect the material decoupled from the variables related to the structure. Nonetheless, this study will have some limitations for extrapolation to 3D structures, as the thermal history of the PLA during the printing of 3D models is different depending on printed geometry, printer parameters, and ambient conditions [20]. However, studying the most fundamental building block of any 3D-printed structure, a 1D filament, will help unveil the phenomenon underlying the changes observed in a 3D-printed e-structure after thermal treatment, simplifying the multivariable problem of 3D-printed parts from a novel perspective not found in the literature.

## 2. Materials and Methods

This study analysed the first three months of the natural ageing of 1D filaments of PLA aged at room temperature and higher temperatures. Our previous study showed that a similar high-molecular-weight PLA required three months of natural ageing at 20 °C in order to stabilise its properties; this is the reason for conducting the ageing studies for up to 91 days. As the primary objective was to compare the samples at room temperature (20 °C) with the samples aged at higher temperatures (39, 51, 65, 75, and 80 °C), no longer times were considered. For a better understanding of the content of this work, a summary of the material production, storage, ageing, and experiments is provided in Figure 1, as well as the research question.

### 2.1. Materials and FFF

The material studied was the 4043D (a commonly studied PLA suitable for 3D printing applications) from Nature Works and supplied by Prusa in the form of filaments with a diameter of 1.75 ± 0.02 mm. The material was printed in a Prusa i3MK3S+ with a 0.4mm nozzle at 215 ± 1 °C (a temperature between 200 and 220 °C, typical printing temperatures for neat PLA and recommended by Prusa) and at an extrusion rate of 20 mm/s, a relatively slow printing speed found in accurate printing processes and studied by other authors [36]. Filaments with diameters between 350 and 450 micrometres were obtained (Figure 1a). This 4043D material was compared with the 2003D from Nature Works, a high-molecular-weight PLA (182.000 g/mol) used in our previous studies [15,17,23]. The material 2003D has been discontinued; thus, we looked for a similar PLA currently used in the literature, and found the 4043D [36]. The PLA 4043D was printed in a room at 23 °C and ambient humidity, then frozen at −24 °C after 2 h of natural ageing at 20 °C to ensure that all the PLA studied were under the same conditions [23], as summarised in Figure 1b.

### 2.2. Ageing Procedure

Printed filaments were stored inside zip bags with a desiccant inside to ensure a low-humidity-controlled atmosphere, essential for avoiding any degradation through the hydrolysis of the PLA. Ageing at room temperature was performed inside these bags, as indicated in Figure 1b. The bags were later placed in a dark room, with no UV light sources and at a controlled temperature of 20 ± 1 °C. The ageing temperatures were 39, 42, 51, 65, 75, and 80 °C. The reason for these temperatures is explained in the discussion, but in summary, they were chosen based on the DSC scan of a 2h aged sample of PLA right after it was printed (Figure 1c) to try to force any potential degradation mechanism inside the ageing device and to probe the importance of adequately ageing the material before making conclusions about it. The ageing at those six temperatures higher than room temperature was performed inside an oil bath, using an "oil bath" device, schematised in Figure 2a. Premium Mineral Oil 10W-40 (Repsol) was used as the liquid bath in the device, as shown in Figure 2b. Then, the PLA filaments were introduced inside PET zip bags with a desiccant inside. After this, PP test tubes were prepared (Figure 2c) by placing a steel ball at the bottom—to ensure that the test tube sinks—and adding 2–3 g of desiccant. Then, the zip bags with the filaments were introduced inside PP test tubes. The PP tubes were sealed with vacuum grease applied to the caps-tube joints (Figure 2c). Finally, the tubes were immersed inside the oil, with the oil previously stabilised at the required temperature. The temperature was monitored with a precision of ±0.1 °C with the help of a thermocouple (FLUKE 50D K/J Thermometer) introduced in a reference test tube—and not directly into the oil—ensuring that the measured temperatures were the actual ageing temperatures. The oil level in the oil bath was slightly below the cap-tube joint, as schematised in Figure 2a. Different methods were tested for the ageing at elevated temperatures, involving water as the hot liquid in the water bath device; however, any method involving water instead of oil failed as the desiccant completely changed its colour in just a few minutes. The cost of the ageing device with the oil and desiccant was EUR 150, making it an economical and practical method for the rest of the researcher community.

### 2.3. Differential Scanning Calorimetry

The differential scanning calorimetry (DSC) was performed on a Mettler Toledo 822e device at a heating rate of 10 °C/min from 30 to 180 °C (Figure 1d). The device was calibrated following the Indium standard. Aluminium crucibles of 40 microlitres were used, in which 4.5 to 5.5 mg of PLA filaments were cut into 2–3 mm pieces and carefully placed inside, ensuring that all the pieces were in contact with the bottom. This was important for minimising measurement errors, as it was found that differences of up to 1 °C were obtained if not all the PLA was in direct contact with the crucible [16].

### 2.4. Tensile Test

Mechanical tests were performed on an Instron 5866 universal tensile test machine with a load cell of 1 kN (Figure 1d). Before the mechanical testing, the samples were attempered at room temperature inside zip bags with a desiccant for 2 h in a room with no sources of sunlight. Then, the same procedure was followed that was described in our previous study for testing the 1D filaments [15,17,23]. The strain rate used was 1 mm/min (consistent with the UNE-EN ISO 527-2:2012) on filaments of 20 mm length, in addition to 7.5 mm extra length on each side glued with a cyanoacrylate-based adhesive to a cardboard frame (Figure 3). Then, the cardboard frame was firmly held with mechanical clamps, as shown in Figure 3b. The clamps were joined to the load cell with rotulas to avoid non-tensile stresses on the filaments during the tests. For modelling mechanical properties, the logistic model proposed in our previous study was used [1].

This is the expression used for the logistic fitting:(1)$$S\left(t\right)=\frac{S_{\infty }\;}{1+{\;\mathrm{Ae}}^{-\mathrm{Bt}}}=\frac{\left(1+A\right)S_{0}\;}{1+{\;\mathrm{Ae}}^{-\mathrm{Bt}}}$$
where $S_{\infty }$, the value of the property at an infinite time of ageing. It corresponds to the stabilised value. It has the units of the property (e.g., stress units such as MPa or GPa for the tensile strength or elastic modulus).A, the ageing potential (adimensional). It describes the gap between the property at zero days (on the material right after it was printed) and infinite days of ageing (stabilized).B, is the ageing rate in units of (1/time), which describes the speed at which the material evolves towards the steady state.This fitting was proven to properly define the evolution of the tensile strength, yield strength and elastic modulus with the natural ageing at room temperature by using just those 3 parameters. The objective was to increase the value of the B parameter, the ageing rate, by increasing the ageing temperature. B was used to quantify the extent to which the ageing rate varied at different ageing temperatures. The logistic fitting is impossible to apply if the material is not ageing, i.e., for the temperatures higher than T_g_ (51 °C and 65 °C). When the logistic fitting was not possible because the material did not evolve with the ageing temperature (51 °C in this study), the linear fitting y = mx + c was used. The 65 °C was not fitted to any curve as there were not enough points.

## 3. Results

### 3.1. Thermal Properties

All the results for the thermal properties measured on the DSC are summarised in Table 1.

It is generally said that the ageing rate of a polymer can be accelerated only at temperatures below T_g_. For this reason, T_g_ was measured on PLA samples right after printing (0.5 and 2 h) to set a maximum ageing temperature. However, how this maximum ageing temperature should be obtained is not further explained. Our goal was to use the highest temperature possible. Two DSC tests were performed to obtain a picture of the T_g_ reaction on the PLA right after it was printed. Figure 4 shows the DSC scans at 10 °C/min of the PLA aged at 20 °C at different times. A phenomenon was found, not described yet in the literature as far as we know, in which two glass transition reactions appeared. These were later sought and found on the PLA 2003D.

In Figure 4, a clear endothermal transition consistent with T_g_, called T_g2_, was identified after 0.5 h of the printed material. A slight endothermal process was noticed before the T_g2_ on the samples aged for 2 h. The experiment was conducted at heating rates of 10 °C/min during the following days (Figure 4). We observed: (1) that the T_g2_ was disappearing, and (2) that the slight endothermal reaction was becoming larger and forming the typical shape of the glass transition; we called it T_g1_ as it was the first glass transition to appear on the DSC heating scan. For the T_g1_ to show up consistently, at least 2 h of natural ageing at 20 °C was required after the FFF process (Figure 4). Samples studied at 0.5 h after being printed did not show T_g1_, making this double T_g_ a potential problem for (a) any DSC in the literature on PLA samples right after printing at ageing times lower than 2 h, (b) for the typically used procedure “first heating – cooling – second heating” [37], as our findings provide information about T_g2_ but not about T_g1_. Notice that the PLA 4043D-based materials are commonly studied in the literature [27,28,29,30,31,32]. Moreover, this T_g1_ phenomenon is very subtle during the first hours. A large mass of PLA-to-surface contact between the PLA and the aluminium crucible made it even more challenging to detect it. This result shows the importance of carefully placing the material inside the DSC crucible in the most controlled and reproducible manner and with a similar mass (not in the range of 2 to 10 mg). Both variables were responsible for shifts of up to +1 °C in the temperature peaks and a widening of the peaks. The evolution of T_g1_ was found to be extremely fast (compare its value at 2 and 3 h on samples aged at 20 °C in Table 1), making it hard to define a maximum temperature limit for accelerated ageing.

After this discussion on the double glass transitions and considering that the T_g_ is defined approximately as the halfway point of the glass transition—depending on the standardised method used—not the glass transition but the onset temperature of the glass transition was used as the maximum temperature at which the ageing could be effectively performed. The onset temperature of the first glass transition (T_g1_) was 42.9 °C (Figure 5a). This temperature was measured from a DSC at 10 °C/min after 2 h of natural ageing at 20 °C, and it is known that for obtaining precise values, slower DSC scans should be used. The heating rate of 10 °C/min is a heating rate that can be easily found in research on PLA from other authors in the literature [30,38,39,40,41,42,43,44] as well as in our previous works [15,17,23]. However, notice the difference between the scan at 10 °C and 3 °C in Figure 5. The onset temperature of T_g1_ on the DSC scan at 3 °C/min was 41.7 °C, decreasing by 1.2 degrees the measured onset temperature.

The evolution of T_g1_ was relatively fast, making it even harder to define a maximum temperature limit for the accelerated ageing. It was found that 42 °C worked very well for accelerating the ageing of the PLA 4043D naturally aged 2 h after FFF (results in Table 1). That temperature was above the measured 41.7 °C, but we decided to use a more conservative temperature to be on the safe side of the daily use of ageing samples 2 h right after printing. The chosen temperature was 39 °C, approximately 3 degrees below the onset temperature, 6 degrees lower than the first glass transition measured (T_g1_ at 10 °C/min on samples aged 2 h at 20 °C after FFF), and 15 degrees lower than the T_g2_ measured at 0.5 h after printing. The 51 °C ageing temperature was chosen as it was close to but below the second glass transition (T_g2_) found on the PLA right after it was printed in order to demonstrate that the T_g2_ measured cannot be directly used as an indicator of the maximum temperature for accelerating the ageing of the PLA, especially if it was measured on a sample right after printing. Temperatures of 65, 75, and 80 °C were used to provide further information on how ageing at higher temperatures could affect the PLA (inside our ageing device) and to show any potential degradation evidence together with the mechanical tests to confirm the safeness of accelerating the ageing of the PLA. The DSC results are discussed together with the mechanical properties.

### 3.2. Mechanical Properties

All the results from the tensile tests on the PLA 4043D are summarised in Figure 6 and Figure 7. Tests at 0.5 h were not viable, as the adhesive clamp needs time to harden, and the minimum possible ageing time at room temperature was 2 h (approximately 0.1 days) from printing to a reliable tensile test.

Thermal and mechanical properties were measured, and they are highly interrelated. For an appropriate discussion, all the results must be considered together. The samples aged at 20 °C were used for setting a reference. In this way, it was possible to compare the evolution of the samples aged at higher temperatures with these controls. As expected, the ageing of these PLA 4043D samples at room temperature was consistent with those in our previous study on a similar PLA, the 2003D. The trends in the evolution of the properties remained similar: (1) the enthalpic relaxation (both temperature and enthalpy) increased until stabilisation; (2) the enthalpic relaxation enthalpy continued to increase even after the stabilisation of the glass transition; (3) the crystallisation and melting reactions remained unaltered; (4) the elastic modulus and yield strength evolved until they stabilised, requiring times similar to the stabilisation of the glass transition. This similar evolution trend was further proven by applying our proposed logistic fitting [17] to the mechanical properties—elastic modulus and yield strength—and the fitting provided an excellent description of the evolution of these two properties.

The logistic fittings in Figure 6 and Figure 7, for both the tensile strength and the elastic modulus, correspond with the following Equations (1) and (2):(2)$$\sigma_{y}\left(\mathrm{MPa}\right)=\frac{62.7}{1+0.13e^{-0.20t}}$$
(3)$$\sigma_{y}\left(\mathrm{MPa}\right)=\frac{3.18}{1+0.13e^{-0.24t}}$$
where $\sigma_{Y}$ is the yield strength in MPa, E is the elastic modulus in GPa and t is the ageing time in days. This confirms that our proposed logistic fittings correctly described the evolution of the mechanical properties with the ageing at room temperature.

The samples aged at 39 °C showed almost stable properties after just one day of ageing, indicating a significant improvement in the ageing rate, the B parameter on the logistic fitting. The logistic fittings were calculated, and Equations (4) and (5) were obtained:(4)$$\sigma_{y}\left(\mathrm{MPa}\right)=\frac{65.4}{1+0.26e^{-3.47t}}$$
(5)$$\sigma_{y}\left(\mathrm{MPa}\right)=\frac{3.41}{1+0.28e^{-2.34t}}$$

In comparing the parameters obtained in the logistic fittings on samples aged at 20 and 39 °C (Table 2), we observed not only that the ageing was accelerated (increase in B), but also that it improved the value of the stable properties (increase in $S_{\infty }$). This result was again consistent with results in the literature, as it was at the limit of the maximum temperature that can be used for accelerating the ageing, as explained beforehand, and it was specifically chosen for this reason. Properties close to the stable ones were obtained after just one day (ageing rates increased by, approximately, 1000 % regarding B values calculated from fitting equations (Table 2)).

It is known, and was here observed, that the T_g_ increases with the ageing of samples after printing. Thus, the maximum ageing temperature could be increased proportionally to the increase in T_g_, but fully optimising the accelerated ageing process was not the objective of this work.

The evolution of the properties to higher $S_{\infty }$ values is explained as follows. The macromolecules with a higher kinetic energy—higher ageing temperature—can surmount higher energy barriers, reach more stable locations, and provide a stronger and more rigid material as well as a higher T_g_ and enthalpic relaxation. This effect was referred to as the summer effect, as samples which might be stable at ambient conditions, if not stored properly, could evolve to higher values due to summer temperatures. The summer effect also applies for explaining the outstanding results obtained on samples aged at 42 °C, in which even higher values were obtained on the DSC scans after just four days for the thermal properties, reaching the T_g1_ 63.3 °C and its enthalpic relaxation 65.6 °C and 5.7 J/g without any signal of degradation.

Samples aged at 51 °C did not evolve, as the temperature was over the T_g1_, proving the importance of studying aged materials. The samples, however, had a relatively high T_g_, as if they were aged. However, the enthalpic relaxation value was as it would be if the material had been just printed, raising a contradiction for any direct relation between the glass transition and the enthalpic relaxation: samples which are not aged regarding the enthalpic relaxation might show T_g_ values similar to those of an aged material. This shows that it is difficult, if not impossible, to correlate some of the thermal properties measured on the PLA. For example, a T_g1_ of 60 °C, an average value of T_g_ found on aged PLA materials, is not necessarily a sign of a high enthalpic relaxation enthalpy (the figure typically used in the literature to indicate how aged the PLA is). However, some trends can be found and modelled [17]. The desiccant remained stable during the first days of ageing, but, after 42 days, it turned into a blueish colour indicative of adsorbed humidity and highlighting a potential hydrolytic degradation of the PLA, Figure 8. No significant signs of hydrolytic degradation were found on the PLA aged at 51 °C on the DSC scans, except for a slight decrease in the crystallisation temperature and an increase in the crystallisation enthalpy. Concerning the mechanical tests, the material behaved mainly as the PLA 2 h aged at 20 °C right after it was printed. The logistic fitting was not possible for these samples, as they did not age at 51 °C. Linear fitting was used instead (y = mx + c), resulting in Equations (6) and (7).
(6)$$\sigma_{y}\left(\mathrm{MPa}\right)=-0.002t+55.5$$
(7)E(GPa)=−0.0023t+2.72
where the meaning of m and x are the slope and the theoretical property at time zero, respectively; $\sigma_{Y}$ is the yield strength in MPa; E is the elastic modulus in GPa; and t is the ageing time in days. It was noticed that the elastic modulus had a slightly decreasing negative slope in Equation (7); note that the relation of -0.0027 relative to 2.23 in Equation (7) is an order of magnitude higher than −0.002 compared with 55.5 in Equation (6). This could be associated with the loss of the flow-induced molecular orientation, which is a known phenomenon that occurs when a polymer is extruded (Figure 9a), due to the shear stress produced in the PLA-nozzle interface (Figure 9b,c) and mainly affects to the elastic modulus [45,46,47]. This phenomenon is more noticeable for small extrusion shapes, such as our filaments, which were in the range of 350 to 450 micrometres. There is almost no literature on the effect of this phenomenon on printed polymers via FFF.

A decrease in the molecular weight—degradation—could also explain the slight decrease in the elastic modulus due to the degradation of the material. This degradation is especially important when the temperature increases over 50 degrees in a humid ambient environment [48]. Although a low-humidity atmosphere was obtained with the different barriers used during the ageing—the oil bath, the PP tube with desiccant, the vacuum grease used in the joint, and the PET zip bag with even more desiccant inside—the desiccant ended up turning blueish after 42 days at 51 °C. As a reference, the desiccant only remained yellow for a few minutes if it was placed under ambient conditions, and if it was placed inside our ageing device, it required some weeks (Figure 8).

Considering both the loss of flow-induced molecular orientation and the degradation as the two possibilities for explaining the decreases in the elastic modulus, the PLA samples were aged and tested at even higher temperatures for one day, forcing the mechanisms of both the degradation and the loss of flow-induced molecular orientation. Note that the influence of the crystallinity was discarded entirely, as the measured values were below 3% for all the samples. Three higher temperatures were analysed: 65, 75, and 80 °C. Both the 75 and the 80 °C temperatures were discarded from the mechanical test, as we observed on the DSC that they crystallised after one day of ageing (up to 28 % of crystallinity content compared with the 2–3% crystallinity contents on the samples aged at lower temperatures), introducing new variables to consider in the system but which are out of the scope of this study. Nonetheless, the desiccant remained bright yellow, as the positive in Figure 9, after one day of ageing at temperatures of 75 and 80 °C. After discarding the temperatures of 75 and 80 °C, the chosen temperature was 65 °C, which did not crystallise during the ageing.

The results from the PLA aged at 65 °C were insufficient to provide information to support any of the mechanisms: (1) it had similar mechanical values to the 42-day-aged PLA at 51 °C, and (2) the error bars of the measurements were within the error bars of the reference samples. The DSC results showed that the material was slightly degraded as both a decrease in the crystallisation temperature and an increase in the crystallisation enthalpies -related to higher mobility of the polymeric chains were detected. After all, the same analysis considering the error bars applies to the samples aged at 51 °C. The elastic modulus decreased by 5–6 % considering the average values, but it is also true that the error bars fell within the error bars of the reference sample. This makes it impossible to conclude why the elastic modulus of the PLA aged at 51 °C decreased. However, the results provide useful information: the degradation of the PLA and the loss of flow-induced mechanical orientation are almost negligible for a sample aged for 42 days at 51 °C, thus making those phenomena completely negligible for samples aged at 39 °C—potentially at 42 °C—for some days.

## 4. Conclusions

With all these data, we can ensure that accelerating the ageing rate of the PLA 4043D at 39 °C for a few days in an oil-bath device is safe, that it effectively ages the PLA, and that it will produce slightly higher properties than if it is slowly aged at room temperature. Furthermore, more critically, no variables related to the 3D structure were mixed in the discussion, making this a potentially essential reference for discussing the effect of heat treatments on 3D-printed parts produced via FFF.

Summarising the results obtained, our answers to our initial research questions are:Questions 1 and 2:

It is safe to accelerate the ageing rate of the PLA at 39 °C for several weeks in our proposed oil-bath ageing device, which aged the material properties close to the stable ones after just one day, but some extra days are needed to fully reach stable properties

Any degradation of the PLA during the ageing for some days at 39 °C can be discarded as it is hardly observable in a sample aged for 42 days at 51 °C.

The mechanical tests, combined with the DSC, were not suitable for decoupling the potential degradation from the loss of flow-induced molecular orientation.

Potentially, higher temperatures could be used with the increase in T_g1_ during the ageing process to obtain faster ageing rates and higher properties due to the so-called summer effect.

Question 3:

The main difference between the samples aged at 39 °C and those aged at 20 °C was the higher stability of the material aged at 39 °C. This has been defined as the summer effect, in which ageing at higher temperatures produces higher stabilisation properties (for the tensile strength, elastic modulus, glass transition, and enthalpic relaxation) due to the higher kinetic energy of the PLA macromolecules that can access more stable locations.

Question 4:Two glass transitions (T_g1_ and T_g2_) were found and characterised which had not been previously described in the literature. For these two glass transitions to be noticed, the printed PLA needed at least 2 h of ageing at 20 °C. This could induce possible mistakes in selecting the ageing temperature, as the 51 °C (below T_g2_) was found to be too high for ageing the material.Samples aged at temperatures above T_g1_ did not age as expected as they increased their T_g_ values whilst remaining at a negligible enthalpic relaxation enthalpy. This makes it impossible to correlate the T_g_ values with those of a material that is or is not aged.The PLA 2003D from our previous studies and the PLA 4043D studied here were found to age similarly and to be relatively similar regarding their thermal and mechanical properties.


This covers the objective of this research: understanding the effects of accelerating the ageing rate on the PLA. We conclude that it can be safely done and that it is worth doing. Future work should use these results to decrease the PLA’s ageing time, with reference to the main difference between the samples aged at an accelerated rate and those studied until now at a standard ageing rate discussed in our previous works. Some questions remain open, such as how to properly decouple the effect of the loss of flow-induced molecular orientation from the effects of the degradation. Furthermore, it would be interesting to see to what extent the PLA properties could be increased by pushing the summer effect to its limits.

Combined with our previous research examining the effects of freezing the PLA to stop its ageing [23], researchers have the option to produce as many materials as they wish in only one day and, under the same conditions, to freeze all the produced material. They can then test the fully aged materials with just a few days of ageing at a higher temperature. This research will provide the research community with data on how the FFF-printed material is affected by these thermal treatments decoupled from all the related structural variables. We consider this a crucial step for properly understanding any 3D-printed structure, though we are also aware of its limitations.

The limitations of this work are that every different geometry that is printed will undergo a different thermal history (for example, a slower cooling rate if the PLA is deposited over a still-hot PLA layer). This might affect the crystallinity contents of the material. This variable was not considered in this study, as the crystallinity percentages were very low and materials with higher contents (samples aged at 75 and 80 °C) were discarded.

## Figures and Tables

**Figure 1 polymers-15-00069-f001:**
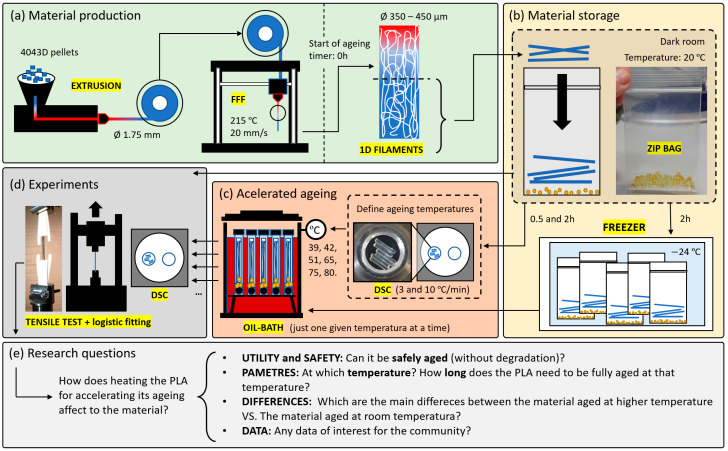
Workflow of the experiments performed: (**a**) production of PLA 1D filaments from pellets; (**b**) storage of PLA filaments; (**c**) accelerated ageing prior to definition of ageing temperatures with a DSC scan on the material right after it was printed; (**d**) experiments performed; (**e**) research questions.

**Figure 2 polymers-15-00069-f002:**
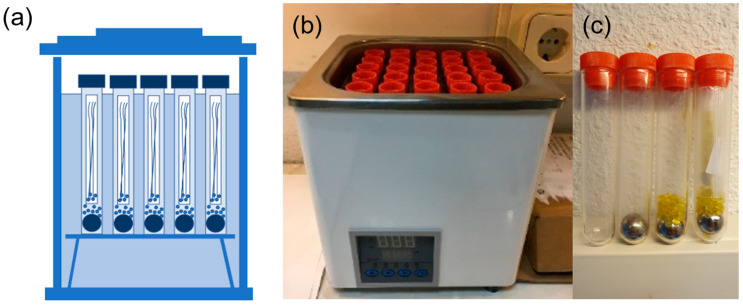
Ageing device. (**a**) Scheme of the ageing device; (**b**) actual picture of the ageing device with 25 test tubes in a 5-by-5 grid, with the middle one used for measuring the temperature; (**c**) preparation of PP test tubes, left to right: PP tube, with a ball of steel, with desiccant, with the samples inside a zip bag with more desiccant inside.

**Figure 3 polymers-15-00069-f003:**
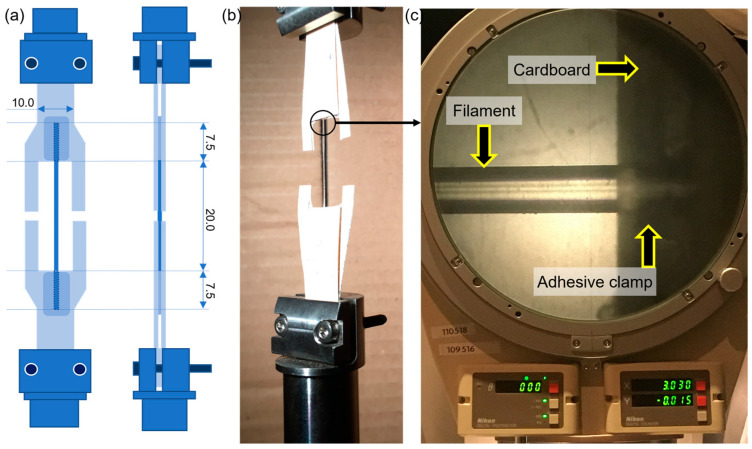
Clamps setup used for the mechanical testing of 1D-printed PLA filaments. (**a**) Scheme of the setup, (**b**) picture of an actual sample prior to testing, (**c**) profilometer detail of the filament, cardboard, and adhesive clamp.

**Figure 4 polymers-15-00069-f004:**
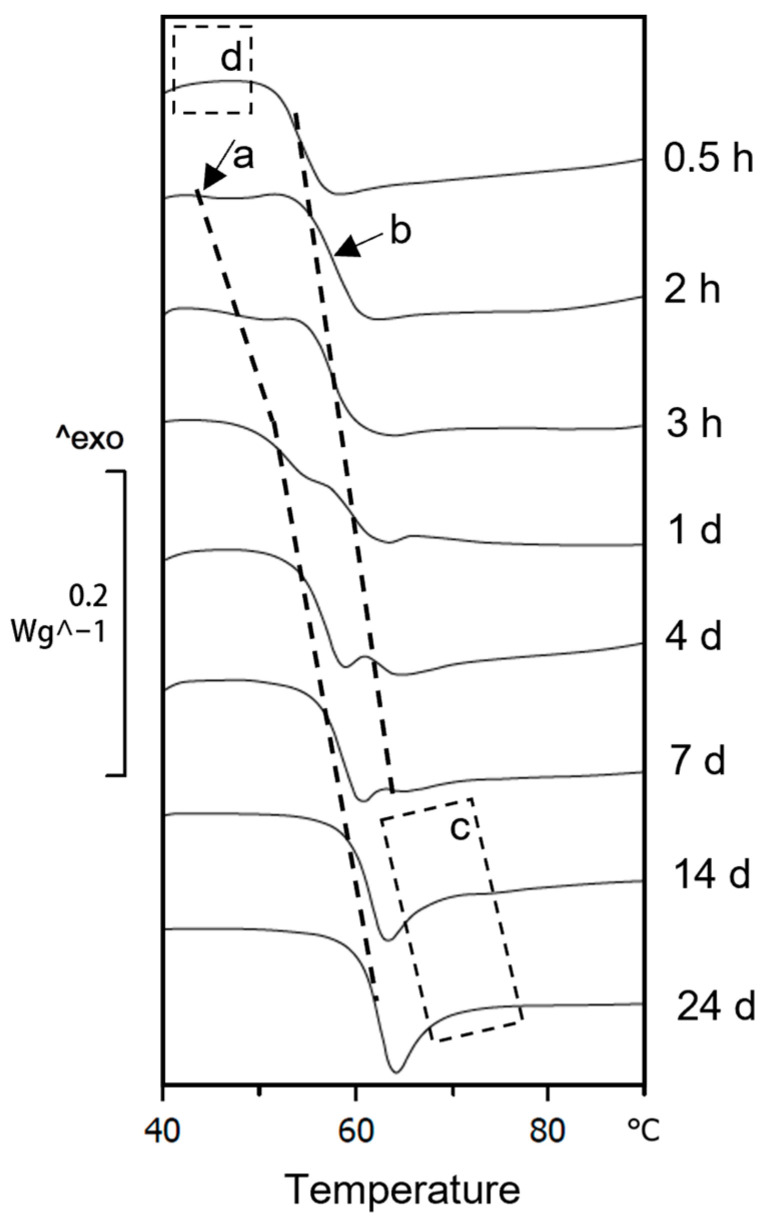
Detail of DSC scans at 10 °C/min of PLA 4043D samples aged at 20 °C, for illustrating the double T_g_ observed: (a) hidden T_g1_ on PLA right after it was printed, which was used to determine the maximum temperature at which the material could be aged and highlight the importance of studying aged materials; (b) second glass transition, T_g2_; (c) missing T_g2_; (d) missing T_g1_.

**Figure 5 polymers-15-00069-f005:**
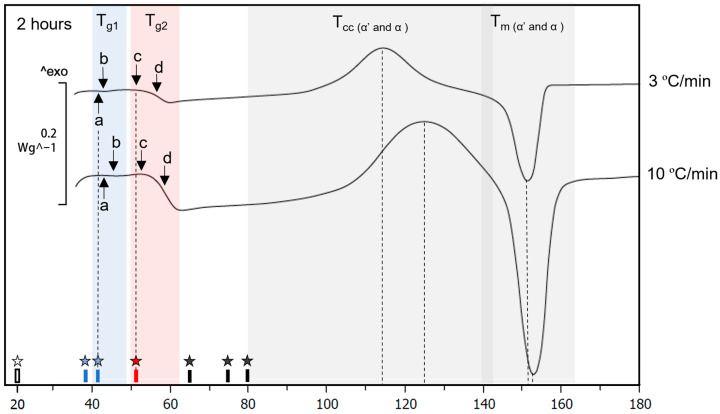
DSC of neat PLA right after printing via FFF at heating rates of 3 and 10 °C/min. Stars indicate the ageing temperatures used: 20, 39, 42, 51, 65, 75, and 80 °C. (a) T.onset of the first glass transition; (b) T_g_ of the first glass transition; (c) T.onset of the second glass transition; (d) T_g_ of the second glass transition. Crystallisation and melting reactions are also indicated. Dashed lines are included for reference.

**Figure 6 polymers-15-00069-f006:**
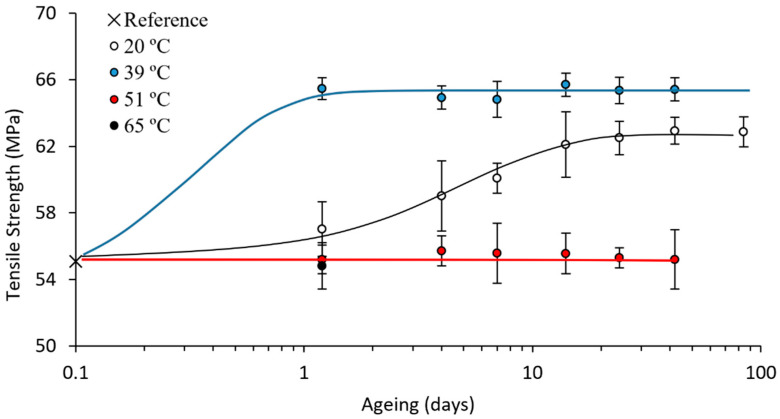
Tensile strength evolution with ageing time at different ageing temperatures. Logistic fittings for 20 and 39 °C, linear fitting for 51 °C. These fittings correspond to Equations (2), (4), and (6), respectively.

**Figure 7 polymers-15-00069-f007:**
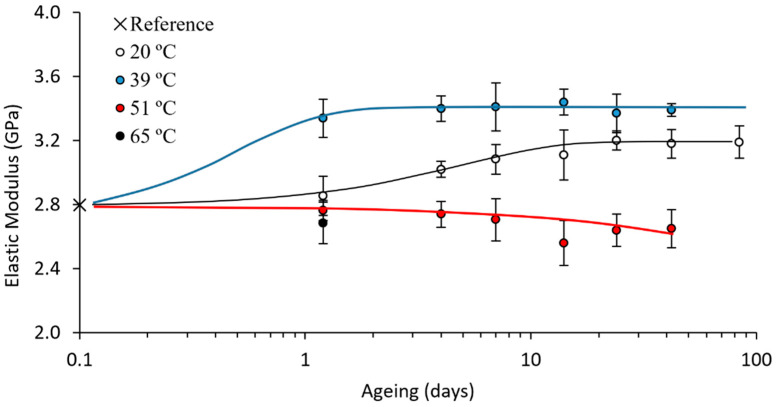
Elastic modulus evolution with ageing time at different ageing temperatures. Logistic fittings for 20 and 39 °C, linear fitting for 51 °C. These fittings correspond to Equations (3), (5), and (7), respectively.

**Figure 8 polymers-15-00069-f008:**
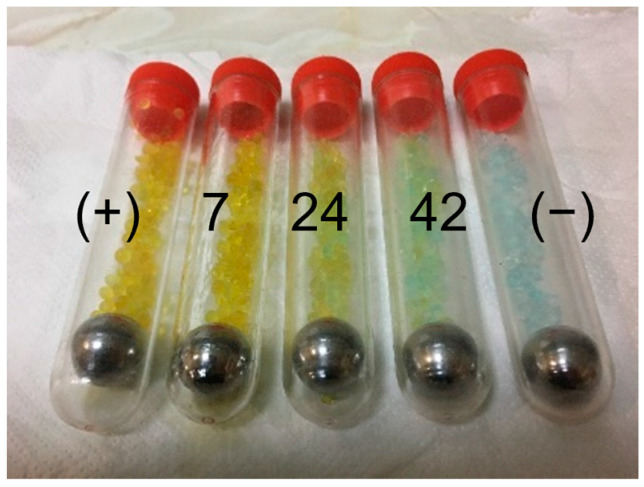
Desiccant control samples after 7, 24, and 42 days of ageing at 51 °C. (+) stands for the positive reference with a desiccant at 0 days, and (−) stands for a negative reference with the desiccant in direct contact with the air for some minutes.

**Figure 9 polymers-15-00069-f009:**
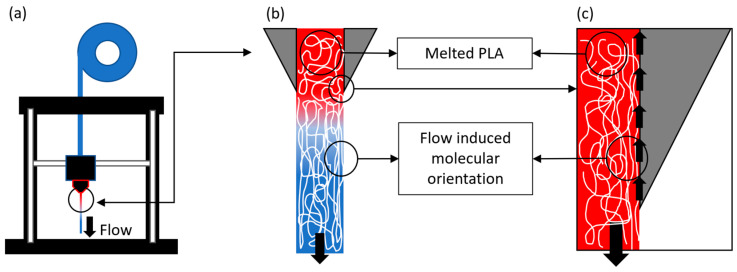
Flow-induced molecular orientation: (**a**) printing process, (**b**) detail of the nozzle and printed PLA, (**c**) detail of the shear forces between the nozzle and the PLA.

**Table 1 polymers-15-00069-t001:** Thermal properties of the 1D-printed PLA. All samples were printed at 215 °C. Temperature measured in °C with a ± 0.5 error (except as indicated), enthalpies in J/g with a 3.4% error of total value following Mettler Toledo guidelines, time in hours (h) or days (d) with an error of ± 0.1h. All aged samples at temperatures above 20 °C were aged after 2 h of natural ageing (for example, one day at 39 °C means that the material has been aged 2 h at 20 °C and one day at 39 °C), crystallinity in percentage (%) with an error of ± 1. Positive enthalpy changes indicate that the material absorbs energy, following the IUPAC convention. T_g.1_ and T_g2_: first and second glass transition on the DSC during heating; T_ER_: enthalpic relaxation temperature; ΔH_ER_: enthalpy of enthalpic relaxation; T_CC_: cold crystallisation temperature; ΔH_CC_: enthalpy of cold crystallisation; T_m_: melting temperature; ΔH_m_: enthalpy of melting; χ_%_: crystallinity percentage. The symbol “-” is used for undetected values.

Ageing	Ageing Temperature	T_g.1_	T_ER.1_	ΔH_ER.1_	T_g.2_	T_CC_	ΔH_CC_	T_m_	ΔH_m_	χ_%_
0.5 h	20 ± 0.1	-	-	-	54.4	123	−15.2	152	15.2	<3
2 h ^1^		45.5	-	-	57.5	123	−15.2	152	15.2	<3
3 h		46.4	-	-	57.6	123	−14.7	151	16.4	<3
1 d		51.5	-	-	59.9	124	−17.0	154	17.2	<3
4 d		54.8	-	-	61.1	121	−16.8	152	16.4	<3
7 d		56.1	58.6	0.15	62.6	122	−15.6	152	15.9	<3
14 d		56.6	58.9	0.79	-	121	−15.8	152	16.3	<3
24 d		56.9	60.0	1.61	-	122	−16.2	152	16.5	<3
49 d		57.2	60.6	2.5	-	122	−15.9	152	16.3	<3
91 d		57.5	61.0	3.7	-	123	−15.7	153	16.5	<3
1 d	39 ± 0.1	58.4	61.4	2.0	-	125	−14.4	153	15.2	<3
2 d		59.0	61.7	3.3	-	124	−17.8	152	18.6	<3
4 d		59.4	62.2	4.0	-	125	−14.5	152	15.7	<3
7 d		60.7	63.2	4.4	-	125	−14.7	152	15.5	<3
14 d		61.0	63.5	4.7	-	124	−15.0	152	15.7	<3
24 d		61.2	63.5	5.0	-	124	−15.9	152	17.2	<3
1 d	42 ± 0.1	59.5	62.1	3.2	-	124	−14.0	152	15.4	<3
4 d		63.3	65.6	5.7	-	124	−15.2	152	16.9	<3
1 d	51 ± 0.1	57.5	61.5	0.6	-	122	−21	152	22	<3
4 d		58.2	62.5	0.9	-	122	−22	152	23	<3
7 d		57.7	62.0	0.8	-	121	−25	151	24	<3
14 d		57.1	63.0	0.7	-	121	−23	152	23	<3
24 d		57.5	62.3	0.8	-	121	−24	152	24	<3
42 d		57.2	62.0	0.8	-	121	−24	152	24	<3
1 d	65 ± 0.1	57.7	61.9	0.6	-	118	−27	150	27	<3
1 d	75 ± 0.1	60.8	-	-	-	- ^2^	- ^2^	154	28	20
1 d	80 ± 0.1	61.9	-	-	-	-	0	153	27	28

^1^ Accelerated ageing materials start from this reference at 2 h of natural ageing at 20 °C. ^2^ A precise calculation is not possible. Values detected, but highly mixed α’- and α-related reactions.

**Table 2 polymers-15-00069-t002:** Comparison of logistic fitting parameters for the yield strength and elastic modulus on samples aged at 20 °C and 39 °C.

Yield Strength	A	B	$S_{\infty }$
20 °C	0.13	0.20	62.7
39 °C	0.26	3.47	65.4
Variation	100%	1635%	4.7%
Elastic modulus	A	B	$S_{\infty }$
20 °C	0.13	0.24	3.18
39 °C	0.28	2.34	3.41
Variation	115%	875%	7.6%

## Data Availability

Not applicable.

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
