# Peer review of "Effects of Accelerating the Ageing of 1D PLA Filaments after Fused Filament Fabrication"

_polymers, 2022, doi:10.3390/polym15010069_

Round 1

Reviewer 1 Report

There are some weaknesses through the manuscript which need improvement. Therefore, the submitted manuscript cannot be accepted for publication in this form, but it has a chance of acceptance after a major revision. My comments and suggestions are as follows:

1- Abstract gives information on the main feature of the performed study, but some details about the experimental tests obtained results must be added. 2- Authors must clarify necessity of the performed research. Research questions, aims and objectives of the study must be clearly mentioned in introduction.
3- The literature study must be enriched. In this respect, authors must read and refer to the following papers: (a) https://doi.org/10.1038/s41598-022-05005-4 (b) https://doi.org/10.1080/1023666X.2021.1892424
4- Authors must explain the limitations of their study.
5- Since it is an experimental investigation, authors must add some real figures (not schematic) to show fabricated specimens and specimens under test conditions. For example, Fig. 3.
6- Why this particular material and proposed method are considered in this study.
7- For FFF process, authors can read and refer recent publications, such as https://doi.org/10.1016/j.polymertesting.2022.107862 and https://doi.org/10.1016/j.tafmec.2022.103317 and other research works.
8- Why these particular temperatures are used for ageing? Is there any standard? Is there any relation to the practical application of the printed parts? And so on.
9- The main reference of each formula must be cited. Moreover, each parameters in equations must be introduced. It seem the manuscript is prepared without care. For example, there is empty bracket in subsection 2.4.
10- Standard deviation is the presented curves must be discussed. In addition, error in calculation must be considered and discussed.
11- In its language layer, the manuscript should be considered for English language editing. There are sentences which have to be rewritten.
12- The conclusion must be more than just a summary of the manuscript. List of references must be updated based on the proposed papers. Please provide all changes by red color in the revised version.

Author Response

REVISOR 1

Thank you so much for such a detailed review. This helps to improve the quality of any published paper.

See PDF manuscript. An expert in the technical English language has checked it.

1.- UPDATED

2.- QUESTIONS, AIMS AND OBJECTIVES HAVE BEEN EXPLICITLY CLARIFIED. Now we provide several reasons through the text for this study:

  1. a) Understand the effect of heating PLA to increase its ageing rate after printing. As we stated in the text, printed filaments have shallow properties that slowly improve with the natural ageing of the material. We have already studied the safety of stopping this ageing by freezing the PLA. Now we want to understand how heating the PLA accelerates its ageing.
  2. b) Because we want to study more filaments and compositions shortly, it is crucial to learn how to accelerate the ageing of the materials until they reach stable properties because we want to have a standard criterion for testing our PLA: we will test them when they are stable. The way of accelerating the ageing is by heating it, and we do try to understand how heating printed PLA samples are affected by this thermal treatment.

-> Degradation? Neglectable, if any, at the temperatures of interest.

-> Loss of flow-induced molecular orientation? Not detected.

-> Stabilized properties? Higher than samples aged at room temperature

-> Ageing rate? Significantly increased.

We do this research by studying 1D printed filaments, decoupling the properties solely related to the material from those related to the structure (raster angle, printing speed, …)

3.- IMPROVED LTIERATURE

  1. a) https://www.nature.com/articles/s41598-022-05005-4 (effect of raster angle and printing speed on 4043D, included)
  2. b) (not sure how it fits)

4.- Limitations: regarding our primary objectives, which are: a) to demonstrate the safeness of accelerating the ageing of the PLA and b) to understand the main differences with the material aged at room temperature, there were no limitations. A secondary objective, like trying to observe the Loss of flow-induced molecular orientation, failed, and we share this for sharing the knowledge.

5.- Figures 1 and 3 have been updated to include images of samples.

6.- Previously, we were working with 2003D. However, the 2003D has been discontinued, and we started to study the PLA that we found the most in the literature, which is the 4043D. Specific data about d-isomer content and molecular weight for both 2003D and 4043D can be found, for example, in this article [1] doi:10.3390/jcs3020052

7.-

  1. a) https://www.sciencedirect.com/science/article/pii/S014294182200383X?via%3Dihub
  2. b) https://www.sciencedirect.com/science/article/abs/pii/S0167844222000696?via%3Dihub

8.- The temperatures were solely based on the DSC diagram of the material at 2 hours after printing. The 39 degrees is a temperature close to the human body. The rest is a typical room temperature in a controlled room, and as explained in the text the higher temperatures, as explained in the text, were chosen to study any potential degradation of the material and to show the importance of not using the value of the second Tg2 which we have not been defined anywhere yet This second Tg could be necessary as de first heating – cooling – second heating, will give information about the second Tg, but not the Tg1.

9.- Added and corrected.

10.- Error has been discussed. At our research institute, up to 10% error (standard deviation) on the mechanical properties is considered ok. It is the main reason for not being able to provide conclusions related to the Loss of flow-induced molecular orientation (a hypothesis on our previous work for explaining the difference in mechanical properties on fully aged samples that were printed at different temperatures). The changes in the mechanical properties are within the error margins, and thus we cannot conclude anything related to the Degradation of the PLA.

11.- Acquired English check.

12.- Rewritten and updated

Changes followed with the “track changes” tool.

Thank you so much for this high-quality review, explaining every aspect to be improved and providing guidance. I do appreciate your effort.

Reviewer 2 Report

I add the comment in the attachment.

Author Response

REVISOR 2

See PDF manuscript. An expert in the technical English language has checked it.

Abstract:

  1. Clarified
  2. Added at the end
  3.  

Introduction:

  1. Corrected
  2. We were referring to our work. Corrected.
  3. Thank you so much for the suggestion. We added it to the text.
  4. Added

Methods:

  1. Added
  2. Extended
  3. Merged
  4. Merged
  5. Added
  6. Added
  7. Merged

Results and discussion:

  1. -
  2. Thank you for the example provided on how to correct it.
  3.  
  4. The logistic fitting is impossible as there are no intermediate points to properly define the curve's shape.
  5. Added

Conclusions:

  1. Rewritten following the expected format.

Thank you so much for this high-quality review, explaining every aspect to be improved and providing guidance. I do appreciate your effort.

Round 2

Reviewer 1 Report

Answer to each question must be presented in details (in response to the reviewer). It seems authors didn't care about the comments, suggested literature and answers. For example, in the answer to the question 3, they mentioned that the first suggested article is cited, but it is not! The same issue for comments in questions 6 and 7. By and large, the manuscript needs a revision considering the previous comments.

Reviewer 2 Report

It is difficult to understand the manuscript since the tracking grammar are not hide. the answer to the reviewer also not satisfied, tend to underestimate the review process, have not serious to answer the review. we can check the review sheet that only said clarified or corrected.

Round 3

Reviewer 2 Report

I attach my comment in the attachment.

Round 4

Reviewer 2 Report

The manuscript is proper to be published.